# Humic Substances Isolated from Recycled Biomass Trigger Jasmonic Acid Biosynthesis and Signalling

**DOI:** 10.3390/plants12173148

**Published:** 2023-09-01

**Authors:** Rakiely M. Silva, Alice N. A. Peres, Lázaro E. P. Peres, Fábio L. Olivares, Sara Sangi, Natália A. Canellas, Riccardo Spaccini, Silvana Cangemi, Luciano P. Canellas

**Affiliations:** 1Núcleo de Desenvolvimento de Insumos Biológicos para Agricultura (NUDIBA), Universidade Estadual do Norte Fluminense Darcy Ribeiro (UENF), Ave Alberto Lamego 2000, Campos dos Goytacazes 28013-602, Brazil; 2Departamento de Ciências Biológicas, Escola Superior de Agricultura ‘‘Luiz de Queiroz’’ (ESALQ), Universidade de São Paulo (USP), Piracicaba 05508-090, Brazil; 3Centro Interdipartimentale di Ricerca CERMANU, Università di Napoli Federico II, Via Università 100, 80055 Portici, Italy

**Keywords:** eco-compatible chemicals, biotic stress, hormonal effect

## Abstract

Intensive agriculture maintains high crop yields through chemical inputs, which are well known for their adverse effects on environmental quality and human health. Innovative technologies are required to reduce the risk generated by the extensive and harmful use of pesticides. The plant biostimulants made from humic substances isolated from recyclable biomass offer an alternative approach to address the need for replacing conventional agrochemicals without compromising the crop yield. The stimulatory effects of humic substances are commonly associated with plant hormones, particularly auxins. However, jasmonic acid (JA) is crucial metabolite in mediating the defence responses and governing plant growth and development. This work aimed to evaluate the changes in the biosynthesis and signalling pathway of JA in tomato seedlings treated with humic acids (HA) isolated from vermicompost. We use the tomato model system cultivar Micro-Tom (MT) harbouring a reporter gene fused to a synthetic promoter that responds to jasmonic acid (JERE::GUS). The transcript levels of genes involved in JA generation and activity were also determined using qRT-PCR. The application of HA promoted plant growth and altered the JA status, as revealed by both GUS and qRT-PCR assays. Both JA enzymatic synthesis (*LOX*, *OPR3*) and JA signalling genes (*JAZ* and *JAR*) were found in higher transcription levels in plants treated with HA. In addition, ethylene (*ETR4*) and auxin (*ARF6*) signalling components were positively modulated by HA, revealing a hormonal cross-talk. Our results prove that the plant defence system linked to JA can be emulated by HA application without growth inhibition.

## 1. Introduction

Conventional agricultural strategies are characterized by fast growing production trends in a short-term period, and weak flexibility of cropping practices limited to the unplanned use of available resources and intensive distribution of synthetic agrochemicals. However, the maintenance of this productive model is currently hampered by both negative externalities related to the impact on environmental quality and human health, to the progressive degradation of soil fertility, and to the long-term loss of crop productivity [1]. The lack of sustainability is amplified by the worsening of adverse cropping conditions, the jeopardizing of food security driven by climate change (soil drought and salinity), and the lowering efficacy and affordability of external inputs (chemical fertilizers, GMO seeds, pesticides, etc.). In this scenario, a widening interest has been devoted to the deployment of more accessible and eco-sustainable technologies based on the application of plant biostimulants (PBs) [2]. PBs are defined as substances and microorganisms, either distinct or combined, that are applied in small amounts, whose function is to stimulate and steer biochemical reactions and physiological activities, purposed to enhance uptake and nutrient use efficiency and improve tolerance to biotic and abiotic stresses while preserving or increasing crop yield and quality [3]. The integration of PBs into sustainable soil management may have a beneficial potential to mitigate climate-change-induced harm, reduce the dependency on chemical inputs and tackle the impact on human health and natural resources [4].

Among the different products, soluble humic extracts are one of the main PB categories applied either as specific application, as mixed formulation or as bio-composite carriers [5]. Currently, the main supplies of commercially available humic products are geochemical sediments, such as coals, lignite, leonardite and peats. In recent years, the exploitation of these non-renewable materials has been under pressure since increasing environmental and ecological concerns have resulted in dedicated regulations limiting their applications [6].

The valorisation of recycled agro-industrial by-products represent a reliable alternative source of bioactive molecules. Organic solid wastes can be converted in to soil amendments, biofertilizers and biostimulants, creating a win–win approach toward more sustainable, profitable and safe agriculture [7]. Vermicomposting technology with earthworm processing is a versatile natural bioreactor for the effective recycling of organic residues into nutritious composts for crop production [8] and a valuable substrate for the isolation of humic substances (HS) with high bioactivity properties [9,10,11].

The stimulatory action of HS is generally attributed to the presence of hormone-like components acting preferentially by the emulation of auxin pathways, due to their remarkable effect on the architecture of radicular systems [12]. However, an extensive range of biochemical and physiological routes have been identified to be affected by humic bioactivity, as outlined by the promotion of shoot development, nutrient uptakes, and crop yield and quality on different soil–plants systems [13]. HS can modify the whole plant’s hormonal balance by modulating a series of primary and secondary metabolic processes related to crop growth and development [14]. In fact, the HS from vermicompost retain and preserve a wide array of molecular components and may therefore deliver targeted or untargeted effects that can strengthen the plant’s adaptation, tolerance and resistance to either pathogens or pedo-climatic and farming or cultivation constraints [15,16]. Therefore, a stringent requirement is a comprehensive understanding of the structural activity correlations between humic components and specific transcriptional, enzymatic or molecular cycles involved in plant responses.

Jasmonic acid (JA) is essential in mediating plant defence responses to stress and regulating growth and development [17]. The acquainted signalling pathways influenced by JA are shaped by plant resistance mechanisms to biotic and abiotic pressures [18]. Moreover, it is well known that jasmonates work as key molecules in the formation of trichrome and that their exogenous application changed the trichrome density of *Arabidopsis* [19] and tomato leaves [20]. Several studies pointed out the subsidiarity effect of humic acids and JA on plants’ tolerance to excess salt and water shortage [21,22,23,24,25,26].

Once subjected to stress, the plants activate phospholipase D (PLDa) in chloroplast membranes to promote the synthesis of linolenic acid [27]. The released linolenic acid (18:3) is a precursor in the JA synthesis through the metabolic conversion into the 13-hydroperoxide derivative *13-HPOT*, which is catalysed by the enzyme *13-LOX* (13-lipoxygenase) [28]. This compound undergoes the action of dehydrogenase forming the first cyclic compound of the route. Moreover, 12-oxo-phytodienoic acid (12-OPDA) is synthesized in the sequence channelled by AOS (alene oxide synthase) and AOC (alene oxide cyclase) enzymes [29]. It was previously reported that plants treated with HS showed high linolenic acid levels combined with the boosting of LOX activity [30,31].

In this study, we investigated the potential elicitor influence of humic acids (HAs) isolated from vermicompost to change the endogenous JA status using the JA-inducible promoter (JERE) fused to the reporter gene *gusA* (JERE: GUS construct) in the tomato model system cultivar Micro-Tom (MT). The composition of the HA was elucidated by Infrared and Nuclear magnetic resonance spectroscopies to provide insight into structural activity relationship. The differential transcription of genes related to the JA biosynthesis and signalling route was measured by qRT-PCR. The effect of HA on leaf trichome density was also evaluated in MT plants. To validate the HA bioactivity the recorded morphological, biochemical and molecular aspects were compared to the outcomes of exogenous JA application.

## 2. Results

### 2.1. Humic Acids Characteristics

The diffuse reflectance infrared Fourier transform (Figure 1a) and 13C-solid-state NMR (Figure 1b) analyses of HA highlighted the main molecular components inherited from the processed organic source. In the DRIFT spectra, the major diagnostic peaks suggested the inclusion of lignin and phenolic derivatives, carbohydrates and lipid compounds. Besides the ubiquitous broad band of O–H stretching at 3310 cm^−1^, the main intense absorption were associate with the following functional groups [32]: C-H stretching (2945 cm^−1^) and bending (1460 cm^−1^) in aliphatic chains, C=O stretching (1660 cm^−1^), C-O stretching, OH deformation (1210 cm^−1^) of COOH in alkyl acids, C=C ring vibration (1590 cm^−1^), C-H deformation (900 cm^−1^) of aromatic structures, and C-O vibrations at 1100 and 1030 cm^−1^ of phenolic and alcoholic OH groups, The specific assignments to organic compounds of DRIFT signals are listed in Table 1.

The ^13^C distribution found in the solid state S-NMR spectrum of HAs confirmed the prevalence of lignocellulose fractions retained in the processed cow manure (Figure 1b). These compounds are identified in the central chemical shift interval by the resonances pertaining to the O-alkyl-C (60–110 ppm) and global aromatic (110–160 ppm) regions, which relatively amounts to up to the 70% rounded of the total spectra area (Table 1). The former NMR section (0–45 ppm) comprises (the CH_2_)n groups of plant- and microbially derived lipid compounds, such as waxes, aliphatic biopolyesters and fatty acids [10,33]. The subsequent sharp and intense peak at 56 ppm depicts the methoxy groups on the aromatic rings of the guaiacyl and syringyl units of the lignin structures of plant woody tissues [11,33]. This chemical shift may also subtend the possible contribution of C-N bonds in peptidic moieties. The O-alkyl-C functions (60–110 ppm) include the C nuclei of monomeric units in oligo and polysaccharide chains. The broad band from 60 to 80 ppm corresponds to the overlapping resonances of the C2, C3, C5 and C6 carbons in the furanoside and pyranoside structures of carbohydrates [10,11,33]. The slight shoulders at 80–85 ppm are derived from the C4 held in glycosidic bond, with the C1 anomeric C identified by the peak shifted at 105 ppm (Figure 1b). The aryl-C region (105–160 ppm) reveals distinct carbon ring resonances in the aromatic, polyphenol and lignin structures. The two broad bands around 115 and 132 ppm indicate the incorporation of apolar rings of complex aromatic derivatives of plant origin. The prominent peak identified at 152 ppm is made by the spectral proximity of various O-substituted ring carbons in lignin and polyphenol [11,33]. The most shifted functional groups shown at 174 ppm are the carbonyl groups of either original molecules, such as aliphatic acids, amino acid moieties, or are derived from composting oxidative processes.

The evaluation of dimensionless structural indicators (Table 1) showed a shared partition between polar and apolar functional groups with a prevalence of hydrophobic features usually found in well-humified processed biomasses [10,11,33]. In addition, the level of the Ar index indicated the significant incorporation of various aromatic components, confirmed by the lower value of the lignin ratio, which was determined by the release in humic extracts of bioavailable phenolic derivatives and of partially decomposed lignin units [11,33].

### 2.2. Plant Phenotypes and Growth

The measurements of MT tomato seedlings revealed distinct modifications produced by MeJA and HA treatments compared to control plants (Figure 2). Differently from the marked root and shoot inhibition induced by MeJA, the application of HA solutions in both concentrations promoted a significant shoot growth (Figure 2A,B), whose fresh weight increased by 35 and 37% concerning no treated plants, respectively. No differences related to control seedlings were observed in the root fresh weight for both HA concentrations.

### 2.3. JERE::GUS Gene Reporter

The *beta-glucuronidase* gene (GUS) fused to the JA response promoter (JERE::GUS) and allowed for the detection of distinguished dye intensity in the MT leaves treated with both Me JA and all concentrations of HA in respect to control plants (Figure 3). The variable higher magnitude of blue colour observed in the HA-treated plants with 4, 8 and 16 mM solutions (Figure 3C–E) indicated the larger but unsteady fostering of JA activity.

### 2.4. JA Synthesis and Signalling

Following the preliminary release of lipids from chloroplast membranes catalysed by phospholipases, the modulation of JA synthesis may be traced by the survey of subsequent enzymatic reactions to discriminate the specific biochemical pattern activated by stress conditions or applied treatment. In the present experiment, we did not observe any relevant alteration in the transcription level of phospholipase PLDa3 found in control plants neither for MeJA nor in HA additions (Figure 4). Conversely, an enhanced transcription rate was shown for the *LOX1.1* enzyme in the MeJA and HA 4 Mm C L^−1^ treatments. The monitoring of subsequent steps of JA biosynthesis did not reveal detectable changes for the involved dehydrogenation nor cyclization reactions catalysed by AOS and AOC enzymatic systems (Figure 4). On the other hand, the connected reductase *OPR3* gene was positively regulated by MeJA and HA, showing a dose–response effect for HA concentrations (Figure 4). This finding is compatible with the GUS staining assay highlighted by the pronounced colouration with higher HA concentration (Figure 3).

Further evaluation of genes transcription encoding for JA factors stressed a more significant level of jasmonate zim domain (*JAZ*) and jasmonic acid-amino acid synthetase (*JAR*) for HA supplies in comparison with MeJA treatment (Figure 5). On the contrary, no differences among treatments and control plants were found for *MYC2*, which belongs to a gene family of F-box transcription factors that are involved in both JA priming activity and gene response to JA (Figure 5).

As expected, the treatments did not modify the transcription level of *CEV1I* (Figure 6). This gene codifies a cellulose synthase and one cell-wall-degrading enzyme expressed in plant JA response under herbivory. However, the genes *ETR* and *ARF*, which are involved in the interaction of JA with other hormones, such as ethylene and auxin, respectively, were positively regulated by HA treatments (Figure 6).

The treatment with MeJA enhanced the type VI glandular trichome density on the adaxial (upper) surface of leaves, while the HAs did not modify the trichome density (Figure 7). The HA reduced the trichome density in the abaxial (lower) leaf surface compared to control plants.

## 3. Discussion

The HS isolated from recycled organic matter may be a viable eco-friendly alternative for plant growth promotion and elicitor of plant resistance to pathogens and environmental pressures [34] depending on the consistency between chemical features and bioactivity of humic products [10,16]. Conformational characteristics and bioactive molecular components are the current acknowledged features underpinning HS’s stimulatory and inducive effects [10,12,14]. The infrared and solid-state NMR analyses of HA from vermicompost revealed the concomitant inclusion of hydrophilic and apolar constituents, which may drive the self-aggregation of dissolved humic colloids in thermodynamic stabilized micelle-like structures [10,11,12,14,15]. Upon interaction with plant tissues, the pliable colloidal humic particles are believed to display complementary behaviours conducive to triggering the bio-active properties in the proximity of plant surfaces. The adhesion to cell membranes, the dynamic rearrangement and unfolding of humic aggregates, are supposed to either trigger the signalling for activation of mild-stress responses and/or convey the release/exchange, towards cell receptors, of carried hormone-like compounds mainly mimicked by soluble carbohydrates, phenolic and lignin components [5,11,12,33]. Among the HA solutions tested in the present work, the intermediate dissolved amounts were identified as the most suitable conditions for plant treatments. The suspended colloidal humic particles are characterized by an optimal critical micellar concentration (CMC) boosting the conformational stability, whose value depends on the combination of hydrophilic and hydrophobic molecular components. As reported in previous experiments [11,33], a dissolved amount, either lower or higher than the CMC value, provides metastable conformations. The unfavourable interactions with the water solution may, respectively, hamper (lower content) the self-aggregation process or promote (higher content) the tightening of assembled molecules that reduce the interaction with cell membranes and favour a colloidal flocculation.

Some of the reaction’s networks in plant defence mechanisms are related to activating JA phytohormone, which regulates a broad spectrum of biological processes, including cell growth and development and feedback responses to biotic and abiotic stresses [35]. Previous reports have shown that humic acids can emulate the action of JA, reducing the effect of stress by stimulating antioxidant enzymes [36]. Furthermore, the observed increase in JA and jasmonate-isoleucine concentration in leaves of plants treated with humic acids was linked with detected changes in auxins, ABA and cytokinin levels, suggesting the modulation of plant hormonal balance [24].

In the present experiment, the HA isolated from vermicompost influenced the JA-modulated plant response in tomato plants. A specific transgenic MT line (JERE::GUS) allowed for the disclosure of all active forms of jasmonate in larger quantities following HA addition, compared to control plants, in staining assays (Figure 3). Even though this assay did not show an evident correlation between dye intensity and applied soluble humic doses, the HA modification of both the JA synthesis and signalling transduction pathway was enlightened by the differential transcription patterns of involved enzymatic systems (Figure 4, Figure 5 and Figure 6).

The first step of JA biosynthesis is the release of α-linolenic acid (α-LeA) from chloroplast galactolipids by phospholipase1 (PLDa1). Both MeJ and HA treatments did not affect the transcription level of *PLDa3*, which codifies enzymes responsible for lipid mobilization from chloroplast membranes. As previously pointed out for the HA induction of α-LeA in plant tissues [28,29], the increase in JA amount in HA-treated plants underlined in staining assay (Figure 3) may lead to the activation of a common enzymatic cascade pathway for the generation of C18 unsaturated fatty acids based on a de novo synthesis from acetyl-coenzyme A (CoA) by the concerted action of acetyl-CoA carboxylase (ACC) and fatty acids synthase (FAS) in the endoplasmic reticulum [37]. 

The oxygenation of α-LeA is the initial step in JA biosynthesis catalysed by lipoxygenases (LOX), and its transcription level was significantly modified by MeJ and HA treatment (Figure 4). The level of LOX is controlled by multiple factors, including a Ca^2+^-dependent control of LOX protein, leading to constitutively elevated JA levels [38]. Ca^2+^ is an early acting second messenger responding to many biotic and abiotic stimuli [39]. The HA can induce Ca^2+^ cytosolic pulse promoting downstream regulation cascade including Ca^2+^-dependent kinase activity [40] and the calmodulin channel [15]. Ca^2+^ is a crucial player in plant responses to environmental stimuli, leading to context-dependent Ca^2+^ fluctuations upstream and downstream of JA biosynthesis or in parallel to JA generation, and is a part of the regulatory network of evolutionary divergent metabolic pathways [41].

Allene oxide cyclase (AOC) is also involved in JA biosynthesis, but both MeJ and HA treatment did not reveal an alteration of its transcription intensity (Figure 4). However, the OPDA reductase3 (*OPR3*) was positively regulated by both MeJ and HA with a dose–response trend according to HA concentration (Figure 4). *OPR3* was permanently used to distinguish between JA- and OPDA-dependent signalling in pathogen countering.

It is conceivable that HA are also involved in the signalling of the JA transduction pathway since the JAZ proteins are connected to Ca^2+^ signalling, JA transcription factors and mitogen-activated protein kinases (*MAPKs*). We observed a relevant variation in the transcription of both *JAR* and *JAZ* genes (Figure 5), while no effect was found for *MYC2* transcription levels in both MeJ and HA treatments (Figure 5).

The JA signalling starts with perception at COI1 acting as an F-box protein [42], which confers hormone specificity [43]. The degradation of *JAZ* allows for the release of positively acting transcription factors, such as *MYC2*, that bind to JA-responsive elements occurring in promoters of JA-responsive genes, thereby initiating the transcription of JA-responsive genes [44]. *MYCs* belong to the bHLH domain-containing TFs and act as both activators and repressors of distinct JA-responsive gene expressions in Arabidopsis [45] and different specific signalling pathways, including the synthesis of auxin [46]. As the events are in cascade and the experimental approach provides a snapshot of dynamic behaviours, it is possible to indicate that the observation time must have influenced the quantification of the MYC transcription level.

The MT phenotypes are modified by MeJ exhibiting root growth inhibition, while HA counteracts the JA action enhancing root growth (Figure 2). MeJ can repress the auxin –*TIR1-AUX/IAA–ARFs* signalling cascade [47], while HA induces the transcription of *ARF6a* (Figure 3). HA can serve as an integrating factor in the auxin–JA interaction, leading to a regulatory loop in sustaining the inducive functions of both auxin and JA signalling (Figure 2 and Figure 3). The candidates responsible for this mediation may be the stimulation of an H^+^-ATPase in the plasma membrane and TOR regulation. Both factors are influenced by auxin-like activity; the target of rapamycin (TOR) is Ser/Thr protein kinase, which in plants is a central hub for complements of different kinds of nutrients, energy, hormone, and environmental stimuli [48]. As a growth hormone, auxin stimulates TOR activity to promote the activation of cell proliferation [49]. One of the most reported effects of HA on plants is the emulation of the auxin effect, such as the promotion of H^+^-ATPase activity [12]. HS can stimulate the expression of the Arabidopsis *TOR* gene [50] and regulate the expression of TOR under nutrient restriction or not [51]. In addition, the mild stress imposed by HA may determine a high level of *SnRK2s* [52] that shut down the TOR-promoted growth thus fortifying the stress adaptation responses [53]. TOR generally intercepts and mediates environmental and nutrient signals to promote growth and development when the nutrient supply is sufficient, whereas SnRK1 acts antagonistically and restores energy homeostasis during stress [54].

The auxin-induced expression of *JAZ1* might supplement the auxin–JA interaction, leading to a regulatory loop in sustaining auxin and JA functionalities [55]. The mechanism of JA–auxin cross-talk also occurs via auxin-dependent transcription factors, like *ARF6* [56], that were positively regulated by both MeJA and HA. The JA signal steering takes place at multiple levels, including *JAR1* and *JAZ* modulation by HA. The capability to bind regulatory elements in the promoter of JA biosynthesis genes leads to a larger JA production, as shown by the staining assay (Figure 3). However, the shoot and root growth were stimulated by HA but not by JA. Taken together, the JA-induced root growth inhibition seems to occur preferentially via the modulation of the effects of auxin/HA in root development since the trichomes were not affected by HA treatment (Figure 7). Glandular trichomes are multicellular and often involved in resistance to insects due to the formation of terpenoids, flavonoids, alkaloids and protective proteins [57]. In addition, they represent a valuable tool for producing secondary metabolites [58]. Genetic evidence for the involvement of JA in glandular trichome formation was obtained by characterizing the tomato homolog of COI1, the central component of JA perception [49]. The link between trichome formation, JA and defence is known and widely reported [57,58]. The HA treatment did not enhance the trichome density in MT leaves, showing further evidence of the complex interrelationship between HA’s induction of AJ biosynthesis and signalling genes and the phenotypic discrepancy evidenced by root growth promotion and low trichome density.

Finally, HA can induce plant secondary metabolisms, as those connected to phenylalanine ammonia-lyase activity and phenolic concentrations [59]. Zaho and colleagues [60], using mutant acp1, showed reduced levels of linolenic acid (18:3) and a corresponding decrease in the abundance of JA. The concomitant increase in JA levels and the recognized stimulus in the metabolism of phenolic compounds induced by HA need to be better understood due to the antagonistic relationship between JA and salicylic acid (SA).

## 4. Materials and Methods

### 4.1. Vermicompost Production

Vermicompost was obtained from cattle manure. The organic residues were mixed, and earthworms (*Eisenia andrei*) were added at a ratio of 5 kg m^−3^ of organic residue. The vermicompost was produced after 90 days of incubation. The average temperature of incubation was 31.5 °C for the first 30 days and 26.2 °C for the next 60 days. The resulting vermicompost exhibited the following characteristics: pH 6.5, C:N ratio of 15:1, 154 g kg^−1^ organic carbon, 10.3 g kg^−1^ nitrogen and 10.5 g kg^−1^ humic acids.

### 4.2. Extraction and Purification of Humic Matter from Vermicompost

Humic acids (HA) from vermicompost were extracted by a 0.5 M NaOH solution. After 24 h under an N_2_ atmosphere, the suspension was centrifuged, and the insoluble residue was separated. The pH of the supernatant was then adjusted to 2.0 using 6 M HCl. Next, the HA was redissolved in reduced distilled water and dialyzed (1000 kDa molecular weight (MW) cut-off) against deionized water. After dialysis, the HA was freeze-dried.

### 4.3. Humic Acids Characteristics

The elemental composition of HA was evaluated using a Perkin Elmer CHN-14800 autoanalyzer. The Diffuse reflectance infrared Fourier transform (DRIFT) spectra of the HA were recorded with a Shimadzu Prestige 21, equipped with a diffuse reflectance accessory, accumulating up to 100 scans with a resolution of 4 cm^−1^. Before DRIFT analysis, the dry samples were finely ground with an agate mortar and diluted with a KBr powder (5/150, *w*/*w*). The solid-state nuclear magnetic resonance spectroscopy was performed with the cross-polarisation magic angle spinning (13C CPMAS-NMR) The spectra were acquired with a Bruker AVANCE 300 NMR spectrometer equipped with a 4 mm Wide Bore MAS probe and operating at a 13C resonating frequency of 75.475 MHz. The powdered HA was spun at 13 ± 1 kHz within a 4 mm Zirconia rotor locked with Kel-F caps A. In total, 4000 scans were collected over an acquisition time of 25 ms and a recycle delay of 2.0 s. The Bruker Topspin 1.3 software was used to collect and analyse the spectra. All free induction decays (FIDs) were transformed by applying a 4 k zero-filling and a line broadening of 100 Hz.

For the interpretation of the 13C-CPMAS-NMR spectra, the overall chemical shift range was split into six regions related to the main organic functional groups: 0–45 ppm (aliphatic-C), 45–60 ppm (methoxyl-C and N-alkyl-C), 60–110 ppm(O-alkyl-C), 110–145 ppm (aromatic-C), 145–160 ppm (O-aryl-C), and 160–190 ppm (carboxyl-C) [24,25]. The relative contribution of each functional group was estimated by relating the area intensity of the corresponding spectral interval to the total spectral area.

The structural properties of the HA were summarized by the evaluation of dimensionless indicators derived from the combination of specific functional groups (11, 33).

The hydrophobic index is the ratio of signal intensities in chemical shift intervals for apolar alkyl and aromatic C components over those of hydrophilic C molecules:

(1) HB/HI = Σ[(0–45) + (45–60)/2 + (110–160)]/Σ[(45–60)/2 + (60–110) + (160–190)].

the alkyl ratio determines the relative contribution of apolar versus polar alyl molecules.

(2) A/OA = (0–45)/(60–110).

The aromaticity index is the total area assigned to aromatic compounds.

(3) Ar = Σ[110–160).

The lignin ratio relates the area of methoxyl-C+N-alkyl groups to that of O-aryl-C.

(4) LigR = (45–60)/(145–160).

### 4.4. Plant Assay

Tomato (*S. lycopersicum* L.) cv. Micro-Tom (MT) harbouring the JA response synthetic element (AGACCGCC) fused to the *gusA* gene was used [59,60]. Non-transgenic MT was also used as a control and is called wild type (WT) in the text. The seeds were surface sterilized with a 3% NaClO commercial bleach for 15 min under agitation. The seeds were washed with sterile water and sown in Petri dishes covered with filter paper moistened in water. The plates were kept in a growth room at 25 °C with a 12 h photoperiod. After germination (4–5 days), the seedlings were treated with increasing concentrations (0, 2, 4, 8, 16.0 and 32 mm C) of HA in glass pots; four replicates (one replicate means five seedlings per pot) were submitted to each treatment. The preliminary assessments on seedling development with variable HA solutions indicated the intermediate (4 mM C) and higher (8 mM C) concentrations as the most reliable doses for the molecular assays on JA synthesis.

JERE::GUS reporter assay: four-day-old JERE::GUS seedlings [61] were treated with HA to detect and compare jasmonic acid-like activity. The tomato seedlings were treated for 4 days with a solution of HA after determining the appropriate concentration for each sample (as described earlier) or with 2 mM CaCl_2_ as a control. Histochemical GUS staining was performed as described previously [62] with minor modifications: incubation time of the seedlings for 1 day in a dark room and using the X-Gluc solution at 37 °C. Seedlings were observed under light microscopy; we searched for the diffuse blue product of the enzymatic reaction. The experiment was set up in a completely randomized design, with 10 replicates per treatment. Representative seedling phenotypes were photographed.

### 4.5. Differential Transcription Level of Genes with RT-qPCR

A sample of 100 mg of fresh leaves *tissues* was homogenised with a mortar and pestle in liquid N_2_. The homogenate was transferred to new RNAse-free microcentrifuge tubes (1.5 mL), and the RNA was extracted using the mini-plant RNeasy Qiagen ® kit (Germantown, USA). Reverse transcription (RT) followed by polymerase chain reaction (PCR) 1 μg of total RNA was used to produce cDNAs. The synthesis was performed using the high-capacity cDNA reverse transcription kit Applied Biosystems, USA). A PCR with a gradient temperature (59, 60 and 61 °C) was performed to confirm the specificity of the primers and the actual melting temperature. Electrophoresis in 2.0% agarose gel with TAE buffer was also performed to confirm PCR products with the specific primers. Primers for the genes *Sl*PDLa3 (Phospholipase D alpha); *Sl*LOX2S (Lipoxygenase 2); *Sl*OPR3 (12-oxiphytodienoate reductase 3); *Sl*AOS2 (Allene oxide synthase 2); *Sl*MYC2 (Transcription factor); *Sl*ARF6a (Auxin response factors); *Sl*ARF8a (Auxin response factors); *Sl*JAR1 (Jasmonoyl-isoleucine synthetase); *Sl*JAZ (Jasmonate ZIM Domain-containing protein); *Sl*ETR4 (Etileno response sensor); *Sl*CEVI57 (Proteinase inhibitor II); *Sl*ACT4 (Actin, endogenous control) were designed with the Primer3 program and their characteristics were evaluated in the Oligothech program, and after a rigorous analysis, they were synthesised by IDT technology (Appendix A). Confirmation of primers specificity was obtained in a high-resolution gel, which gave single PCR products at the different temperatures tested and with the expected size. The melting curve performed in StepOne ™ System (Thermo Fisher Scientific, Waltham, MA, USA) also confirmed specificity. The Real-time PCR (RT-qPCR): for statistical validation, two independent tests in the thermal cycler StepOne™ System, with mRNA extracted from the independent experiments, were performed.

### 4.6. Measurement of Trichome Density and Number

The terminal leaflet of the first pair of leaves formed after the beginning of the treatment application (5th and 6th leaf from cotyledon) was dissected along the longitudinal axis into 15 × 3 mm strips covering the entire leaf blade (avoiding the primary veins) and fixed on microscope slides with transparent enamel. 10 individuals per treatment were sampled, and four different ranges were analysed per plant (n = 40) for each side (abaxial/adaxial). Photographs were taken using a Leica S8AP0 loupe (Wetzlar, Germany) set to 80×, coupled to a Leica DFC295 camera (Wetzlar, Germany). Trichome counts were performed on the images, and side views of the images were used to allow for the correct classification of trichome types. Trichome densities were then calculated as trichome counts per unit leaf area.

## 5. Conclusions

We proved using a gene-reporter staining assay that HA from vermicompost modified the JA content in tomato plants. The qRT-PCR analysis showed that both JA synthesis and signalling were positively affected by the application of HA solutions resulting in larger transcription than in control plants. However, phenotypes traits, such as root growth and trichome density, in leaves were divergent from JA activity, suggesting a complex integration with other plant regulators also induced by HA, as observed in this study with auxin response factor (ARF6). These findings are ground-breaking contributions to enforce the knowledge and use of HA-based biostimulant from local available recycled biomasses as accessible technologies to strengthen the plant adaptation and responses to biotic and abiotic stress conditions. The comprehensive understanding of structural activity relationship is an unavoidable requirement to conceive the tailored exploitation of humic materials. The update of scientific insights on the metabolic regulation of soil–plant resilience mechanisms represents a valuable repository support for the reliable adoption of sustainable, affordable and safety farming practices.

## Figures and Tables

**Figure 1 plants-12-03148-f001:**
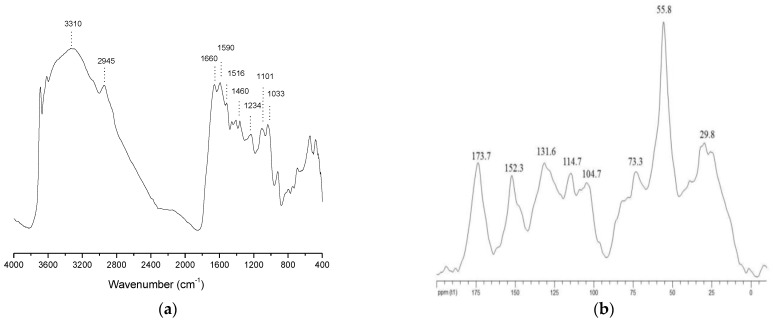
Spectrum of IR-TF (**a**) and CP/MAS 13C NMR (**b**) of humic acids isolated from vermicompost.

**Figure 2 plants-12-03148-f002:**
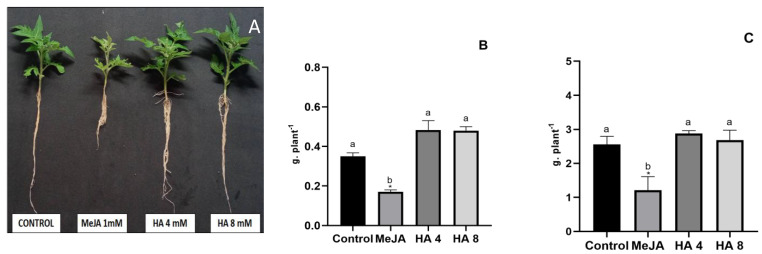
(**A**) Phenotypes of 30-day-old MT tomatoes seedlings exposed to water as a control, 1 mM MeJA, 4 and 8 mM C HA (humic acids). Fresh weight of roots (**B**) and shoots (**C**) treated with 1 mM MeJA, 4 and 8 mM C HA. Data represent the mean and bars standard deviation *(n* = 10). Means followed by different letters are significantly different by the LSD test (*p* < 0.05).

**Figure 3 plants-12-03148-f003:**
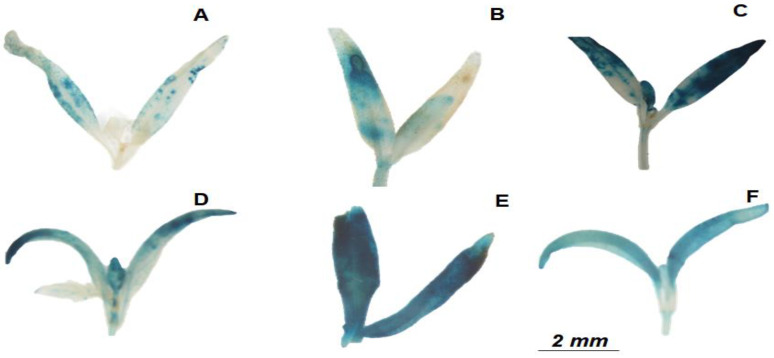
Leaves of Micro-Tom (MT) with gene reporter JERE: GUS were used to perform the GUS staining assays: (**A**) control, (**B**) 1 mM MeJ, (**C**) HA 4 mM C L^−1^, (**D**) HA 8 mM C L^−1^, (**E**) HA 16 mM C L^−1^, (**F**) HA 32 mM C L^−1^.

**Figure 4 plants-12-03148-f004:**
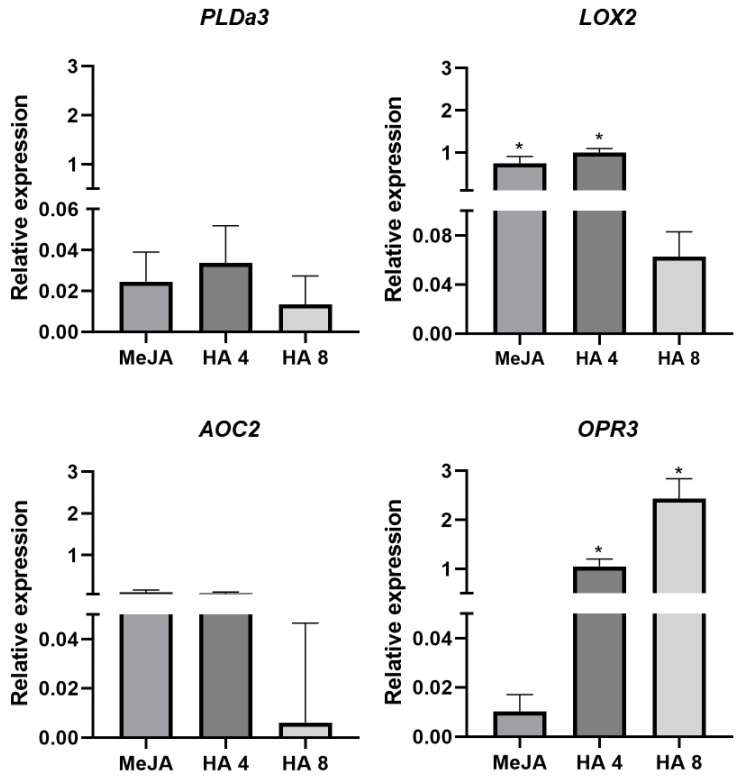
Gene expression linked to jasmonate biosynthesis. phospholipase1 (*PLDa1*), lipoxygenase (*LOX2*), alene oxide cyclase (*AOC*), 12-oxo-phytodienoic acid (*OPDA*) reductase3 (*OPR3*) genes in MT tomatoes treated with 1 mM of methyl jasmonate (MeJA) and 4 and 8 mM C of humic acids (HA) isolated from vermicompost. Total RNA was extracted from leaves and subjected to real-time qPCR analysis. Data represent the mean of three independent samples with SD. * significant difference at *p* < 0.05 by *t* test. The data are expressed concerning control treatment considered = 0.

**Figure 5 plants-12-03148-f005:**
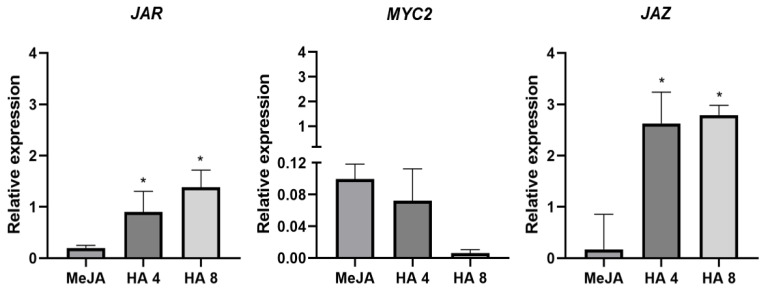
Gene expression linked to jasmonate signalling. bHLH transcription Factor (*MYC2*) jasmonate zim domain (*JAZ*) and jasmonic acid-amino acid synthetase (*JAR*) in tomato MT exogenous treated with 1 mM methyl jasmonate (MeJA) and 4 and 8 mM C of humic acids (HA) isolated from vermicompost. Total RNA was extracted from leaves and subjected to real-time qPCR analysis. Data represent the mean of three independent samples with SD. * significant difference at *p* < 0.05 by *t* test. The data are expressed concerning control treatment considered = 0.

**Figure 6 plants-12-03148-f006:**
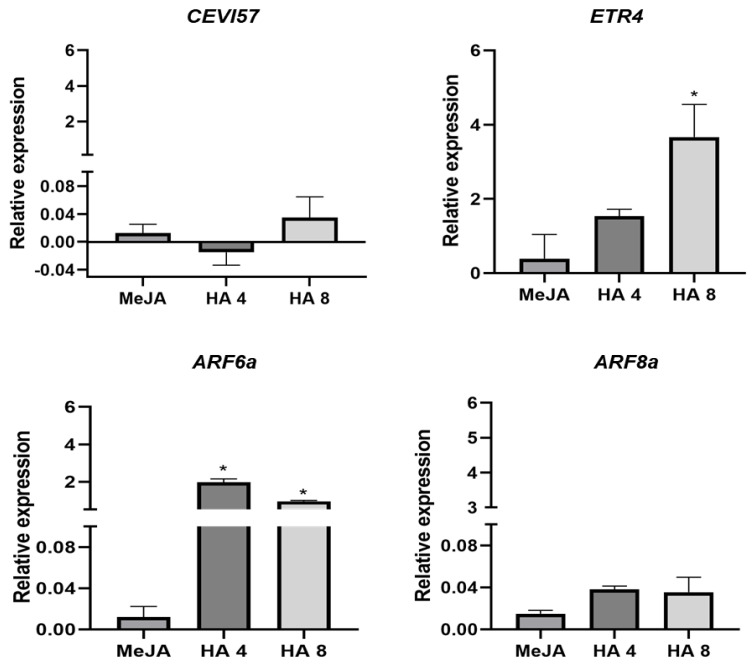
Gene expression linked to herbivory (cellulose synthase-*CEVI57*), ethylene-jasmonate cross-talk (ethylene response sensor-*ETR4*) and jasmonate-auxin cross-talk (auxins response factors, ARF6 and ARF8) in tomato MT exogenous treated with 1 mM methyl jasmonate (MeJA) and 4 and 8 mM C of humic acids (HA) isolated from vermicompost. Total RNA was extracted from leaves and subjected to real-time qPCR analysis. Data represent the mean of three independent samples with SD. * significant difference at *p* < 0.05 by *t* test. The data are expressed concerning control treatment considered = 0.

**Figure 7 plants-12-03148-f007:**
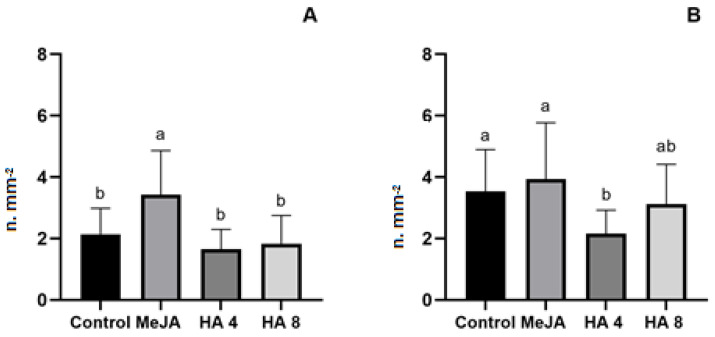
Effect of methyl jasmonate (MeJA) and 4 and 8 mM C of humic acids (HA) isolated from vermicompost on glandular trichome density on MT tomato leaves. (**A**) mean density of type VI glandular trichome on the adaxial (upper) leaf surface and (**B**) abaxial (lower) l leaf surface. N = 10 plants per treatment, one leaf examined per plant. Densities are calculated from counts of trichomes on two leaf disks per leaf. Data represent the mean and bar standard deviation. Means followed by different letters are significantly different from the LSD test (*p* < 0.05).

**Table 1 plants-12-03148-t001:** Molecular composition of HAsdetermined by spectroscopy analyses. (a) Peak assignments (cm^−1^) in DRIFT spectra. (b) Relative distribution (%) of the signal area over chemical shift regions (ppm) in ^13^C-CPMAS-NMR and derived structural indexes.

a DRIFT (cm^−1^)									
2945	1660	1590	1460	1260		1100	1030		
CH_2_ alkyl chains	COOH fatty acids	C=C aromatic units	CH2 alkyl chains	C-O alkyl acids/aromatic ehter		C-O phenols/polysaccahrides	C-O carbohydrates		
b ^13^C CPMAS NMR (ppm)									
C=O 190-160	O-aryl-C 160-145	Aromatic-C 145-110	O-alkyl-C 110-60	CH_3_O/CN 60-45	Alkyl-C 45-0	HB	A/OA	Ar	LR
6.5	8.4	20.3	28.3	16.7	19.8	1.3	0.7	28.7	2.0

## Data Availability

The data may be provided depending on specific demand.

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
