# Peer review of "Humic Substances Isolated from Recycled Biomass Trigger Jasmonic Acid Biosynthesis and Signalling"

_plants, 2023, doi:10.3390/plants12173148_

Round 1
Reviewer 1 Report
The current study aims to to determine Humic substances extracted from recycled biomass stimulate jasmonic acid synthesis and signaling. The present manuscript does not provide the systematic approach, lucid presentation, and critical assessment needed to introduce newcomers to the field. I have some major issues with the content (experimental design, statistical analysis etc.), which make the manuscript unacceptable for publication in the current format.
The MS It is poorly written, with grammatical and formatting errors throughout the MS, which frequently prevents a thorough understanding of the logic described. Therefore, I suggested authors to remove grammatical and formatting errors from their manuscript before peer review for reconsideration of their work for consideration in this journal. Additionally, the English language also needs to be improved (which frequently prevents a thorough understanding of the logic described) throughout the manuscript, but does not detract from the science.
Author Response
Thank you for taking the time to provide feedback on the manuscript. Your input is greatly appreciated. We made the required changes but disagreed with the criticism of the experimental design and statistical analysis.
We send the ms to language revision, convert the text to the British language and fix the problems with cited references.
Reviewer 2 Report
see attachment

Author Response
Thank you very much for your comments, they were particularly useful for improving the manuscript. Here are the answers to the main questions raised by the reviewer.
- The density of trichome type VI was measure as morphological descriptors of AJ effect. In lines 86-88 we describe that “jasmonates work as key signaling molecules in the formation of trichrome and their exogenous application changed the trichrome density of Arabidopsis [31] and tomato leaves [32]”. We try to access biochemical (stain assay), molecular (level of transcription of genes that codify proteins relate to biosynthesis and AJ signalization) and phenotype trials (trichome) related to JA and HA.
- We use the template of ms submission to embed the Fig 1. The Quality Control did not detect any problems, but we will forward this question to the professional staff responsible for layout and final editing.
- The x axis of Figs 4-7 is dimensionless i.e. relative expression as described in the Fig caption “The data are expressed concerning control treatment considered = 0”
- The references were unified according to Plants norms.
Reviewer 3 Report
The submitted manuscript addresses the current issue of the use of biologically active substances in agriculture as a substitute for chemical substances. The study of the influence of humic substances created as a product of vermicompost can still be considered current. The text is written relatively carefully with logical continuity. It is not entirely clear from the text why the density of trichome was monitored. Why was this indicator selected? Are there correlational relationships between monitored parameters? If so, please complete them. The methodology is adequate. The results, especially in the trichome analysis section, could be more descriptive. The discussion is sometimes descriptive. I recommend enlarging fig 1. Furthermore, for figs 4–7, the description of the x axis would be added. It is clear from the description of the graph that these are variants of the experiment. It is necessary to unify the citation of the used literary sources.
Author Response
Reviewer #3
The authors assessed the impact of humic acids (HA) from vermicompost on the biosynthesis and signaling pathway of JA in tomato seedlings. Through qRT-PCR analysis, they observed that HA treatment influenced the transcript levels of genes involved in JA and auxin biosynthesis and signaling. However, the research design appears inadequate to fully support their hypothesis. The variety of methods employed might not be entirely robust. Despite these limitations, the paper is well-written and easy to follow. Below are some additional comments:
- If these are genes or operons, the first letter is in lowercase. On the contrary, proteins go without italics. Please revise all cases throughout the manuscript and figures.
Answer: Thanks for the comments. We revised the annotation.
- Line 30-31, based on the limited results such as gene expression, I don't think your results provide evidence of any defense phenotype linked to HA application.
Answer: the conclusion summarized in lines 30-31 provides evidence that applying HA increases the JA on plants, enhancing genes that codify JA synthesis and signalling.
- Line 78-88, is there any evidence that HA-related derivatives can participate in JA biosynthesis or JA signaling?
Answer: In lines 76-77, we expose that humic acids and JA can reduce osmotic stress (references 19, 20, 21, 22, 23] and plant growth and development [reference 24], followed by a brief JA pathway showing previous works that reported enhance of JA precursors induced by HA and level of JA in plants treated with AH.
- Line 148 and Figure 2, what about the results of 4Mm and 8 mM MeJA and 1Mm HA treatments? Then we can compare these different treatments. And which concentration of HA can display the maximum inducement of plant growth?
Answer: We did not understand the issue raised. The MeJA (methyl Jasmonate 1 mM) and HA (humic acids, 4 and 8 mM C – carbon).
- Figure 3. The same question: which concentration of HA can display the maximum inducement of gene reporter JERE?
Answer: No clear relationship was found between HA concentration and Gus colouration. Despite some methodological aspects not allowing a stoichiometric ratio between JA receptor and GUS colouring, plant scientists know well and accept that the intensity of GUS colour is proportional to hormone level.
- Did you try to collect plant samples treated with HA and analyze the JA levels? If you want to prove that HA can trigger JA biosynthesis, this experiment is necessary. Gene expression is not enough.
Answer: We know that gene expression is insufficient and use physiological and biochemical markers to express changes in all JA forms in plant tissues using a staining assay.
- Line 78-88, Line 262-270. Please refer to the paper published in MPMI, by Zhao, Zhenzhen, et al. "Involvement of Arabidopsis acyl carrier protein 1 in PAMP-triggered immunity." Molecular Plant-Microbe Interactions 35.8 (2022): 681-693. They found that the gene ACP1 regulates JA biosynthesis by influencing the accumulation of 18:3 fatty acids. The reduced JA level plays a significant role in plant defense. Therefore, the relationship between HA and JA, as well as the potential defense responses, needs further clarification. I believe designing more experiments would help to elucidate these connections and support your opinions.
Answer: we insert the recommended citation in the new version of the ms.
Round 2
Reviewer 1 Report
I did not see any response letter from authors?
I did not see any response letter from authors?
Author Response
R "The current study aims to to determine Humic substances extracted
from recycled biomass stimulate jasmonic acid synthesis and signaling.
The present manuscript does not provide the systematic approach, lucid
presentation, and critical assessment needed to introduce newcomers to
the field. I have some major issues with the content (experimental
design, statistical analysis etc.), which make the manuscript
unacceptable for publication in the current format.
Comments on the Quality of English Language
The MS It is poorly written, with grammatical and formatting errors
throughout the MS, which frequently prevents a thorough understanding of
the logic described. Therefore, I suggested authors to remove
grammatical and formatting errors from their manuscript before peer
review for reconsideration of their work for consideration in this
journal. Additionally, the English language also needs to be improved
(which frequently prevents a thorough understanding of the logic
described) throughout the manuscript, but does not detract from the
science"
A The 1st Reviewer made hard criticism on the manuscript about the
critical assessment, the innovation and on the methodological approaches
(experimental design etc.
Although we do not agree with this general comments, no detailed list of
specific points for the various section of the manuscript have been
provided by the reviewer; it is hence difficult o to provide suitable
response to addreess the scientific issues raised by the Reviewer
Globally, in the revised version the text was extensively revised with
respct to grammar, sintax, editing and style. We performed an accurate
revision of the different sections (Introduction, Result, Discussion ,
Conclusion and M6M) to improve the description, the readibility and the
unserstanding of background, approaches and outputs
Please refer to the uploaded version with tracked cu
hanges
Reviewer 2 Report
Thank you for your feedback on the minor revisions made to the manuscript. However, I want to address that two reviewers have indicated the need for major revisions. This suggests that further enhancements are expected in order to meet the desired standards.
Could you please provide point-by-point response to reviewers' comments? This approach will help streamline the communication process and ensure that my responses are effectively aligned with the specific comments provided.
And the same, based on the limited results such as gene expression, I don’t think your results provide evidence of any defense phenotype linked to HA application.
Thank you for your feedback on the minor revisions made to the manuscript. However, I want to address that two reviewers have indicated the need for major revisions. This suggests that further enhancements are expected in order to meet the desired standards.
Could you please provide point-by-point response to reviewers' comments? This approach will help streamline the communication process and ensure that my responses are effectively aligned with the specific comments provided.
And the same, based on the limited results such as gene expression, I don’t think your results provide evidence of any defense phenotype linked to HA application.
Author Response
Re#2
Comments and Suggestions for Authors
The submitted manuscript addresses the current issue of the use of biologically active substances in agriculture as a substitute for chemical substances. The study of the influence of humic substances created as a product of vermicompost can still be considered current. The text is written relatively carefully with logical continuity. It is not entirely clear from the text why the density of trichome was monitored. Why was this indicator selected? Are there correlational relationships between monitored parameters? If so, please complete them. The methodology is adequate. The results, especially in the trichome analysis section, could be more descriptive. The discussion is sometimes descriptive. I recommend enlarging fig 1. Furthermore, for figs 4–7, the description of the x axis would be added. It is clear from the description of the graph that these are variants of the experiment. It is necessary to unify the citation of the used literary sources.
Thank you very much for your comments, they were particularly useful for improving the manuscript. Here are the answers to the main questions raised by the reviewer.
- The density of trichome type VI was measure as morphological descriptors of AJ effect. In lines 86-88 we describe that “jasmonates work as key signaling molecules in the formation of trichrome and their exogenous application changed the trichrome density of Arabidopsis [31] and tomato leaves [32]”. We try to access biochemical (stain assay), molecular (level of transcription of genes that codify proteins relate to biosynthesis and AJ signalization) and phenotype trials (trichome) related to JA and HA.
- We use the template of ms submission to embed the Fig 1. The Quality Control did not detect any problems, but we will forward this question to the professional staff responsible for layout and final editing.
- The x axis of Figs 4-7 is dimensionless i.e. relative expression as described in the Fig caption “The data are expressed concerning control treatment considered = 0”
- The references were unified according to Plants norms.